# The Alignment Problem in Curriculum Learning

## Abstract

In curriculum learning, teaching involves cooperative selection of sequences of data via plans to facilitate efficient and effective learning. One-off cooperative selection of data has been mathematically formalized as entropy-regularized optimal transport and the limiting behavior of myopic sequential interactions has been analyzed, both yielding theoretical and practical guarantees. We recast sequential cooperation with curriculum planning in a reinforcement learning framework and analyze performance mathematically and by simulation. We prove that infinite length plans are equivalent to not planning under certain assumptions on the method of planning, and isolate instances where monotonicity and hence convergence in the limit hold, as well as cases where it does not. We also demonstrate through simulations that argmax data selection is the same across planning horizons and demonstrate problem-dependent sensitivity of learning to the teacher's planning horizon. Thus, we find that planning ahead yields efficiency at the cost of effectiveness. This failure of alignment is illustrated in particular with grid world examples in which the teacher must attempt to steer the learner away from a particular location in order to reach the desired grid square. We conclude with implications and directions for efficient and effective curricula.

## 1 Introduction

Advances in AI and machine learning enable the possibility that artificial systems may autonomously facilitate human goals, including human learning. Design of such systems requires addressing a value alignment problem (Russell, 2019; Christian, 2020), which requires interacting with the system to achieve the desired goals. Toward this end, formalizing models of cooperation among agents that bridge human and machine learning is an important direction for research. In this paper, we identify a novel value alignment problem in the context of agents that facilitate learning, and we identify and test sufficient conditions for ensuring value alignment for curriculum learning.

Learning may be facilitated by the teacher planning ahead, which becomes a problem of reinforcement learning. There exists an extensive literature on curriculum learning (Elman, 1993; Khan et al., 2011; Pentina et al., 2015; Matiisen et al., 2020; Graves et al., 2017); however, this literature focuses on naive learners rather than those that reason cooperatively about the teacher's selection of data. Theoretical results are limited (Milli and Dragan, 2020) and have not systematically considered the possibility of alignment problems or their solutions. Recent advances in theoretical foundations for cooperative inference admit a more unified formal treatment (Wang et al., 2020b), which is necessary to understand whether, when, and why alignment problems arise.

We formalize the alignment problem in curriculum learning via the mathematical condition of consistency. Given a teacher and learner cooperatively communicating, the teacher aims to convey a distribution $\theta$ on the finite set of possible hypotheses $\mathcal{H}$ to the learner, over an infinite horizon. That is, if $\theta_n$ denotes the learner's distribution at the $n$-th round of communication, the alignment problem is to have $\lim_{n \to \infty} \sum_{h \in \mathcal{H}} |\theta(h) - \theta_n(h)| = 0$. When the teacher is conveying a specific hypothesis $h'$ to the learner, the distribution to be learned is $\theta = \delta_{h'}$, a Dirac distribution.

We investigate the alignment problem in curriculum learning by recasting sequential cooperative Bayesian inference (SCBI) as a Markov decision process (MDP). In doing so, we retain the theoretical strengths of prior

formalizations which yielded proofs of consistency and rates of convergence, while considering the benefits and drawbacks of planning by the teacher. Section 2 gives background on one-off cooperative inference and sequential cooperative inference, as well as the interpretation of SCBI as a Markov chain. Section 3 recasts SCBI as a Markov decision process, distinct from the trivial realization of a Markov chain as an MDP, contrasts SCBI with no planning ahead vs. using a teaching plan which calculates several steps into the future, and isolates the theoretical basis of misalignment. One main result of this section is that for a class of reward/cost functions, planning infinitely far ahead is equivalent to not planning at all; this includes using a reward proportional to the probability of selecting the correct hypothesis. The other main result is to give a condition for monotonicity in expectation which yields a sufficient requirement for alignment. Section 4 gives examples and simulations. Section 5 gives related work, and Section 6 concludes.

**Notation.** $M \in \mathbb{R}_{>0}^{|\mathcal{D}| \times |\mathcal{H}|}$ is the joint distribution for the teacher and learner between data and hypotheses, with $|\mathcal{D}|$ many rows and $|\mathcal{H}|$ many columns, with $M_{(d,h)}$ the entry of the joint distribution corresponding to datum $d$ and hypothesis $h$. $M^{\theta,\lambda}$ is the joint distribution $M$ normalized using Sinkhorn scaling to have column sums equal to $\theta$ and row sums equal to $\lambda$. That is, for every $h \in \mathcal{H}$, $\sum_{d \in \mathcal{D}} M_{(d,h)}^{\theta,\lambda} = \theta(h)$ and for every $d \in \mathcal{D}$, $\sum_{h \in \mathcal{H}} M_{(d,h)}^{\theta,\lambda} = \lambda(d)$. $\pi : \mathcal{P}(\mathcal{H}) \to \mathcal{P}(\mathcal{D})$ is a teaching strategy used by the teacher for a single round of teaching, while $\pi_R^N : \mathcal{P}(\mathcal{H}) \to \mathcal{P}(\mathcal{D})$ is the teaching strategy obtained from $\pi$ by planning $N$ teaching rounds into the future and using the random variable $R$, representing rewards/costs inherent to the problem. $\delta_{M_{(d,-)}^{\theta,\lambda}/\lambda(d)}$ is the atomic distribution on $\mathcal{P}(\mathcal{H})$ with atom $M_{(d,-)}^{\theta,\lambda}/\lambda(d)$; i.e. $\delta_{M_{(d,-)}^{\theta,\lambda}/\lambda(d)} \in \mathcal{P}(\mathcal{P}(\mathcal{H}))$, the space of distributions on the distributions on hypotheses, where the Markov operator $\Psi_\pi$ in our formalism is acting. $\Psi_\pi^N$ denotes $\Psi_\pi : \mathcal{P}(\mathcal{P}(\mathcal{H})) \to \mathcal{P}(\mathcal{P}(\mathcal{H}))$ composed with itself $N$ times. Frequently we will shorten the notation $\delta_{M_{(d,-)}^{\theta,\lambda}/\lambda(d)}$ to $\delta_{(d)}$.

## 2 Background

Curriculum learning involves selecting a sequence of learning problems that lead the learner to a desired knowledge or capability. We formalize these as a sequence of data that lead the learner to a target hypothesis. Throughout, we will assume teachers and learners are probabilistic agents reasoning over discrete and finite spaces of hypotheses $h \in \mathcal{H}$ and data $d \in \mathcal{D}$. Recall, in standard probabilistic inference, learners will update their posterior beliefs $P(h|d)$ in proportion to the product of the prior beliefs, $P(h)$ and the likelihood of the data, $P(d|h)$, as dictated by Bayes rule.

**One-off cooperative inference.** Cooperative inference between probabilistic agents differs from standard Bayesian inference in the second agent, the teacher, who selects the data, and in that the agents reason about each other's beliefs. Based on the shared joint distribution between data and hypotheses, the teacher reasons about the learner's beliefs, and samples data to pass according to the learner's current distribution on hypotheses, the joint distribution, and the desired hypothesis to be conveyed. The learner then reasons based upon what data they have been passed by the teacher and infers, based on the shared joint distribution, what hypothesis the teacher is attempting to convey. This process may be represented mathematically by the following system of equations:

$$P_L(h|d) = \frac{P_T(d|h)P_{L_0}(h)}{P_L(d)}, \qquad P_T(d|h) = \frac{P_L(h|d)P_{T_0}(d)}{P_T(h)} \tag{1}$$

where $P_L(h|d)$ represents the learner's posterior probability for hypothesis $h$ given datum $d$; $P_T(d|h)$ is the probability of the teacher selecting datum $d$ to convey hypothesis $h$; $P_{L_0}(h)$ represents the learner's prior for hypothesis $h$; $P_{T_0}(d)$ is the teacher's prior for selecting data $d$; $P_L(d)$ and $P_T(h)$ are normalizing constants. Sinkhorn scaling of matrices (i.e. alternating row-column normalization of the joint distribution) may be used to solve equation (1), and the result is an optimal entropy-regularized plan for transporting beliefs (Wang et al., 2019; 2020b).

**Sequential cooperative inference.** In sequential cooperative Bayesian inference (SCBI), a teacher and learner participate in rounds of learning. To convey a particular hypothesis (or belief on the space of possible hypotheses) from the hypothesis-space $\mathcal{H}$, in each round the teacher passes a datum $d \in \mathcal{D}$ to the learner, and the learner updates their belief distribution accordingly. At the end of each round, the teacher and learner

both update their posterior distributions to become their prior distributions in the next round (Wang et al., 2020a). Each round of learning behaves as in cooperative inference, where the system of equations (1) must be solved. However, at the end of each round, each round differs in having updated the prior, which is one marginal constraint for Sinkhorn scaling in (1).

The process of teaching-learning-updating works as follows: Beginning with the joint distribution $M_n$ of the previous round and distribution $\theta_n \in \mathcal{P}(\mathcal{H})$, which represents the learner's beliefs from the previous round of teaching-learning, where $\mathcal{P}(\mathcal{H})$ is the simplex of probability distributions on $\mathcal{H}$, the teacher computes the Sinkhorn scaling of $M_n$ with row sums $\lambda$ and column sums $\theta_n$. Call this $M_{n+1}$. Here $\lambda$ is an underlying distribution on $\mathcal{D}$ reflecting inherent biases in selecting particular data points; $\lambda$ is typically taken to be the uniform distribution. Then the teacher uses the distribution $M_{n+1}\hat{\theta}$ to sample datum $d_{n+1}$ from $\mathcal{D}$ and passes it to the learner, where $\hat{\theta}$ is the desired belief on hypotheses which the teacher wishes to convey, typically a Dirac distribution, corresponding to a corner of the simplex. The learner then calculates $M_{n+1}$ in exactly the same way as the teacher, then multiplies $\theta_n$ by the likelihood of selecting $d_{n+1}$. Normalizing gives a distribution $\theta_{n+1}$. The process then repeats inductively, with $n$ replaced everywhere by $n+1$.

**SCBI as a Markov chain.** The process of SCBI can be realized as a Markov chain on $\mathcal{P}(\mathcal{H})$ (Wang et al., 2020a). With $T_d : \mathcal{P}(\mathcal{H}) \to \mathcal{P}(\mathcal{H})$ the map bringing the learner's prior to posterior when data $d$ is chosen by the teacher; and $\tau : \mathcal{P}(\mathcal{H}) \to \mathcal{P}(\mathcal{D})$ the map of the teacher's sample distribution based on the learner's prior, and $\tau_d$ the $d$-th component of this map, the Markov transition operator for a fixed hypothesis $h \in \mathcal{H}$, $\Psi(h) : \mathcal{P}(\mathcal{P}(\mathcal{H})) \to \mathcal{P}(\mathcal{P}(\mathcal{H}))$ is defined as:

$$(\Psi(h)(\mu))(E) := \int_E \sum_{d \in \mathcal{D}} \tau_d(T_d^{-1}(\theta)\mathrm{d}(T_d^*(\mu))(\theta) \tag{2}$$

where $T_d^*$ is the push-forward of $T_d$ on Borel measures[1], $\mu$ is a Borel probability measure on the simplex $\mathcal{P}(\mathcal{H})$, and $E \subseteq \mathcal{P}(\mathcal{H})$ is a Borel measurable subset. $T_d$ and $\tau$ are computed as above by using the Sinkhorn scaling of the joint distribution between data and hypotheses.

Wang et al. (2020b) consider the problem of a teacher and learner communicating cooperatively in discrete rounds of teaching/learning. The teacher and learner reason using Bayesian inference at each round, without any reference to what future probability distributions on the available hypotheses might be, and without reference to any costs/rewards the teacher uses in order to determine what distribution to use to sample the data which they are passing to the learner. Although there is a discussion of SCBI as a Markov chain in (Wang et al., 2020b), there is no extension of the method to a Markov decision process. Here we extend the formalism to include planning ahead by the teacher, as well as rewards/costs the teacher uses in planning in order to bias the learner towards/away from a particular hypothesis.

## 3 Curriculum planning via Markov decision processes

*Curriculum planning* involves the teacher planning several teaching moves in advance. We may model this using a Markov decision process (MDP) as follows: Let the *state space* of the process be given by $\mathcal{S} = \mathcal{P}(\mathcal{H})$, the probability distributions on $\mathcal{H}$. Let the *action space* of the MDP be given by $\mathcal{A} = \mathcal{D}$, the data available for the teacher to pass to the learner (interpretation: an action $d \in \mathcal{D}$ corresponds to passing $d$ to the learner). We fix an underlying distribution $\lambda \in \mathcal{P}(\mathcal{D})$, which represents any inherent bias towards selecting or not selecting some particular data. In SCBI (Wang et al., 2020a), $\lambda$ is the uniform distribution. We will allow the reward function $R$ to be a combination of positive and negative pieces (to include costs, if some hypotheses are particularly undesirable).

The transition probability $T(\omega|d, \theta)$ of the MDP between probability distributions $\omega$ and $\theta$ on $\mathcal{H}$, based upon the teacher selecting datum $d$, is $\lambda(d)$ if $\omega = T_d(\theta)$ and is zero otherwise. A teaching strategy then consists of a plan $\pi : \mathcal{P}(\mathcal{H}) \to \mathcal{P}(\mathcal{D})$, effectively, 'sample datum $d$ using $\pi(\theta)$ when the current distribution on hypotheses is $\theta$.' In SCBI, the $d-th$ component of the plan $\pi$ is $\pi_d(\theta) = \tau_d(\theta)$, i.e. the adjustment of the teacher's distribution according to the learner's current distribution. The teacher's strategy can be made

---

[1] i.e. $T_d^*(\mu)(E) = \mu(T_d^{-1}(E))$.

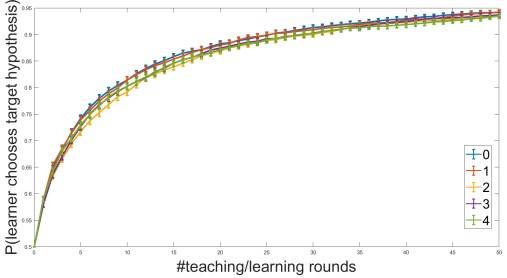 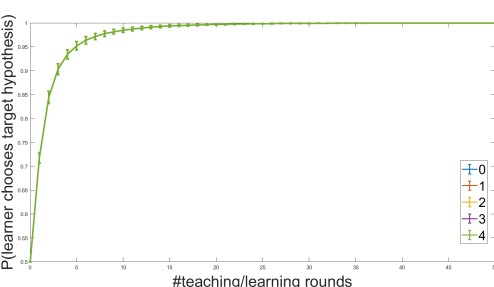

Figure 1: Analysis of convergence to the target hypothesis as a function of longer curricula (0-4) for probabilistic action selection and argmax action selection, given 3 data and 2 hypotheses. We show the probability of the target hypothesis according to the learner versus # teaching/learning rounds, using $X(\theta) = \theta([\text{true hypothesis}])$. **(left)** On average, when the teacher selects actions probabilistically (i.e. Eq. 3), longer curricula yield a marginal difference. **(right)** On average, when the teacher selects actions using argmax, longer curricula is not different from not planning ahead. All curves for 0 steps ahead to 4 steps ahead are exactly overlaid.

deterministic if $\pi_d$ is an atomic distribution with a single atom for every $d \in \mathcal{D}$. Throughout this paper we will be focusing on plans $\pi$ which amount to the teacher preparing a curriculum by calculating what might happen in future rounds of learning based upon the current data selection.

Explicitly, the plan $\pi_R^N : \mathcal{P}(\mathcal{H}) \to \mathcal{P}(\mathcal{D})$ corresponding to planning ahead $N$ moves based on the teaching strategy $\pi$ and a random variable $X$ is realized by the following procedure:

$$\pi_R^N(\theta)(d) = \text{Norm}\left(M_{(d,h)}^{\theta,\lambda}\mathbb{E}_{\Psi_\pi^N \delta_{(d)}}[R] + C\right) \tag{3}$$

Here $\text{Norm}(\cdot)$ is normalization to ensure that $\sum_d (\pi_R^N(\theta)(d)) = 1$, $M^{\theta,\lambda}$ is the Sinkhorn scaling of $M$ so that for every $d \in \mathcal{D}$ and $h \in \mathcal{H}$, $\sum_{d' \in \mathcal{D}} M_{(d',h)}^{\theta,\lambda} = \theta(h)$ and $\sum_{h' \in \mathcal{H}} M_{(d,h')}^{\theta,\lambda} = \lambda(d)$. The $h$ in Equation (3) is the target hypothesis the teacher desires to convey, $R : \mathcal{P}(\mathcal{H}) \to \mathbb{R}$ is a random variable representing a reward or cost, and $\Psi_\pi : \mathcal{P}(\mathcal{P}(\mathcal{H})) \to \mathcal{P}(\mathcal{P}(\mathcal{H}))$ is the Markov operator induced by the teaching plan $\pi$, i.e. for a Borel measure $\mu \in \mathcal{P}(\mathcal{P}(\mathcal{H}))$ and $E \subseteq \mathcal{P}(\mathcal{H})$ a measurable subset,

$$\Psi_\pi(\mu)(E) = \int_E \sum_{d \in \mathcal{D}} \pi_d(T_d^{-1}(\theta))d(T_d^*\mu)(\theta), \tag{4}$$

and $C$ is a constant so that what is being normalized to a distribution is non-negative (typically chosen to be $\epsilon + \min\left\{M_{(d,h)}^{\theta,\lambda}\mathbb{E}_{\Psi_\pi^N \delta_{(d)}}[R] | d \in \mathcal{D}, h \in \mathcal{H}\right\}$, for some small $\epsilon$). Frequently we will drop the subscript $R$, to simplify notation.

Note that the behavior of the teaching/learning interaction will vary, depending upon the normalization constant $C$. On one extreme, as $C \to 0^+$, $\pi_R^N(\theta)(d) \to \text{Norm}\left(M_{(d,h)}^{\theta,\lambda}\mathbb{E}_{\Psi_\pi^N \delta_{(d)}}[R]\right)$. Because of the potential non-positivity, but the overall normalization, this corresponds to a signed probability measure. At the other extreme, as $C \to \infty$, $\pi_R^N(\theta)(d) \to \frac{1}{|\mathcal{D}|}$, independently of $\theta$; i.e. the teacher's distribution is the uniform distribution on $|\mathcal{D}|$, regardless of the learner's beliefs, hence the teacher's choice of data is random. In particular, if $C$ is much greater than the expectation term for some particular choice of data $d$, then $\pi_R^N(\theta) \approx \frac{1}{|\mathcal{D}|}$. In order to make the distribution positive, we must have:

$$C > \left| \min_{d \text{ s.t. } M_{(d,h)}^{\theta,\lambda}\mathbb{E}_{\delta_{(d)}}[R] \leq 0} M_{(d,h)}^{\theta,\lambda}\mathbb{E}_{\Psi_\pi^N \delta_{(d)}}[R] \right| \tag{5}$$

On the other hand, in order to preserve the teacher's knowledge and avoid random data selection, we would also like:

$$C \leq \min_{d | M_{(d,h)}^{\theta,\lambda}\mathbb{E}_{\Psi_\pi^N \delta_{(d)}}[R] > 0} M_{(d,h)}^{\theta,\lambda}\mathbb{E}_{\Psi_\pi^N \delta_{(d)}}[R] \tag{6}$$

However, for random variables $R$ which place too high a cost on some outcomes, versus too small a reward, it may be impossible to simultaneously meet these two conditions. That is, ensuring a positive probability distribution may create a more uniform distribution for the data which already have a positive probability of selection, prior to the addition of the normalization constant $C$.

In Equation (3), if $R(\omega) = \omega(h)$, this corresponds to the teacher calculating $N$ steps ahead in order to choose the data at the current step which will increase the likelihood the learner will be able to infer the target hypothesis in the future. Furthermore, we can replace the expectation of the random variable $\omega(h)$ with an expectation that accounts for rewards/costs inherent to the problem. For example, if $\pi$ is the SCBI teaching strategy, then $\pi(\theta)(d) = \frac{M^{\theta,\lambda}_{(d,h)}}{\theta(h)}$, where $h$ is the target hypothesis (Wang et al., 2020b).

The expectation term of (3) may be simplified as follows, letting $\tilde{\theta} = M^{\theta,\lambda}_{(d,-)}$, and assuming that the original teaching plan follows the scheme of SCBI:

$$\mathbb{E}_{\Psi^N \delta_{\tilde{\theta}}}[R(\theta)] = \sum_{d_1,\ldots,d_N \in \mathcal{D}} \tau_{d_N}(\tilde{\theta}) \tau_{d_{N-1}}(T_{d_N}(\tilde{\theta})) \cdots \tau_{d_1}(T_{d_2} \circ T_{d_3} \circ \cdots \circ T_{d_N}(\tilde{\theta})) R(T_{d_1} \circ \cdots \circ T_{d_N}(\tilde{\theta})). \quad (7)$$

This formula will be useful for simulations, as it allows us to replace the integral arising from the expectation $\mathbb{E}_{\Psi^N_\pi \delta_{(d)}}[R]$ by a finite (though growing exponentially in the number of steps planning ahead) sum. The teacher, when planning a curriculum consisting of finitely many data points, can only guide the learner to a finite number of distributions on hypotheses.

### 3.1 Optimal policy

We may compute the optimal policy as follows:

$$\mathbb{E}\left[\sum_{t=0}^{\infty} \gamma^t R_{a_t}(\theta_t, \theta_{t+1})\right] = \sum_{t=0}^{\infty} \gamma^t R(\theta_t) P_{a_t}(\theta_t, \theta_{t+1}) = \sum_{t=0}^{\infty} \gamma^t R(\theta_t) \lambda(d(t)) \delta_{\theta_{t+1}, T_{d(t)}(\theta_t)} \pi_{d(t)}(\theta_t) \quad (8)$$

Assuming that $\lambda$ is uniform, i.e. the teacher has no underlying bias with regards to the data, and also using the fact that $\delta_{\theta_{t+1}, T_{d(t)}(\theta_t)}$ is only nonzero when $T_{d(t)}(\theta_t) = \theta_{t+1}$, this becomes:

$$= \frac{1}{|\mathcal{D}|} \sum_{t=0}^{\infty} \gamma^t R\left(T_{d(t-1)} \circ T_{d(t-2)} \circ \cdots \circ T_{d(0)}(\theta_0)\right) \pi_{d(t)}\left(T_{d(t-1)} \circ T_{d(t-2)} \circ \cdots \circ T_{d(0)}(\theta_0)\right) \cdot$$

$$\cdot \pi_{d(t-1)}\left(T_{d(t-2)} \circ \cdots \circ T_{d(0)}(\theta_0)\right) \cdots \pi_{d(0)}(\theta_0)$$

The optimal policy which maximizes the expectation term in Equation 8 is therefore the argmax over all possible functions $d : \mathbb{N} \to \mathcal{D}$, where the action at step $t$ is $d(t-1)$, and argmax over functions $\pi : \mathcal{P}(\mathcal{H}) \to \mathcal{P}(\mathcal{D})$. Note that Equation 8 is the same as Equation 7 with a factor of $\frac{1}{\mathcal{D}}$ in front, $N \to \infty$, reverse indexing, and a discount factor $\gamma \in [0,1]$. In particular, this implies that taking an argmax over the distribution in Equation 3 gives a finite horizon approximation of the optimal policy.

We may state this more formally as follows:

**Theorem 1** *The optimal policy is given by the following:*

$$\lim_{\gamma \to 1} argmax_\pi argmax_{d:\mathbb{N} \to \mathcal{D}} \mathbb{E}\left[\sum_{t=0}^{\infty} \gamma^t R(\theta_t)\right] = argmax_\pi \lim_{N \to \infty} argmax_{d:\{0,\ldots,N-1\} \to \mathcal{D}} \pi^N(\theta_0) \quad (9)$$

**Proof:** Note that normalization does not affect which entries of $\pi^N(\theta)$ are larger than others, so we have:

$$\lim_{N \to \infty} argmax_{d:\{0,\ldots,N-1\} \to \mathcal{D}} \pi^N_{d(N-1)}(\theta_0) = \lim_{N \to \infty} argmax_{d:\{0,\ldots,N-1\} \to \mathcal{D}} \left(M^{\theta,\lambda}_{(d(N-1),h)} \mathbb{E}_{\Psi^N_\pi \delta_{(d)}}[R] + C\right)$$

$$= \lim_{N \to \infty} argmax_{d:\{0,\ldots,N-1\} \to \mathcal{D}} \left(M^{\theta,\lambda}_{(d(N-1),h)} \mathbb{E}_{\Psi^N_\pi \delta_{(d(N-1))}}[R]\right)$$

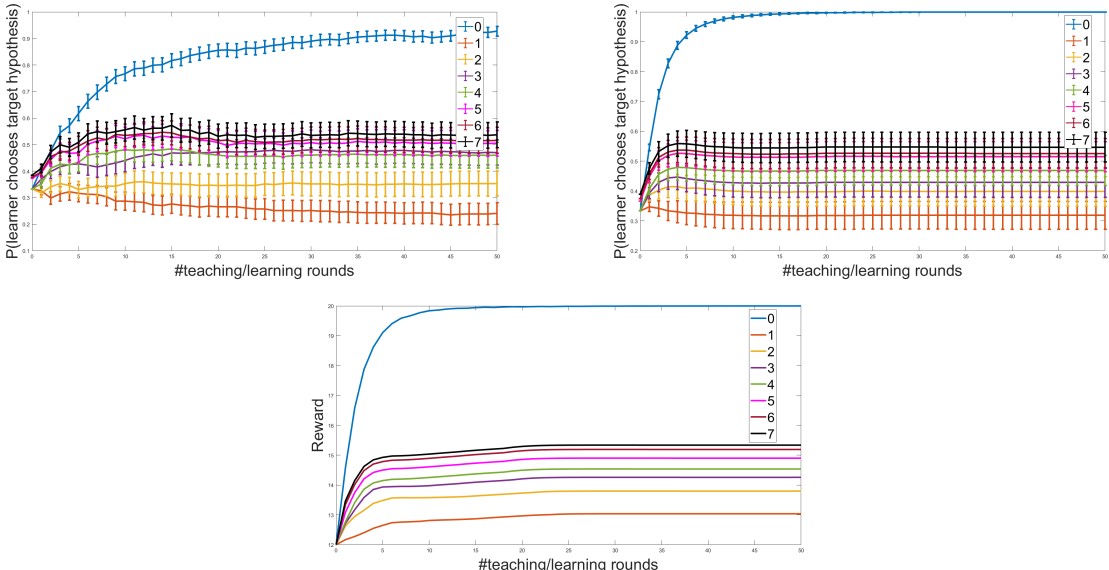

Figure 2: Analysis of convergence as before using reward $R(\theta) = 10 \cdot d_{L^1}(\theta, \delta_{h_{\text{bad}}}) - d_{L^1}(\theta, \delta_{h_{\text{true}}})$. On average, when the teacher selects actions probabilistically **(upper left)** or deterministically (argmax) **(upper right)** curricula yield worse outcomes. The average reward is shown in **(lower)**. Note that the reward increases in the number of steps planning ahead ($> 1$ steps), and slowly approaches the reward of no planning. Planning ahead leads to misalignment between learners and teachers, when the teacher attempts to avoid an undesirable hypothesis. Note that further planning ahead reduces misalignment, consistent with the fact that planning sufficiently far ahead is equivalent to not planning at all.

$$= \lim_{N \to \infty} \operatorname{argmax}_{d:\{0,\dots,N-1\} \to \mathcal{D}} \left( T_{d(N-1)}(\theta)(h) \mathbb{E}_{\Psi_\pi^N \delta_{(d(N-1))}}[R] \right)$$

Because $T_{d(N-1)}(\theta)(h)$ is multiplied by every term of the expectation sum, we obtain that the above is equal to:

$$= \lim_{N \to \infty} \operatorname{argmax}_{d:\{0,\dots,N-1\} \to \mathcal{D}} \left( \mathbb{E}_{\Psi_\pi^N \delta_{(d(N-1))}}[R] \right) = \lim_{\gamma \to 1} \mathbb{E} \left[ \sum_{t=0}^{\infty} \gamma^t R(\theta_t) \right]$$

Taking the argmax over policies $\pi : \mathcal{P}(\mathcal{H}) \to \mathcal{P}(\mathcal{D})$ then yields the result. $\square$

The policy $\pi_{d(t)}(\theta_0)$ which satisfies the argmax above at time step $t$ should be the Dirac distribution $\delta_{d(t)}$, where $d(t)$ is the optimal choice of datum at time $t$. We may therefore obtain an estimate of the optimal policy by taking the argmax of $\pi^N$ as $N$ becomes progressively larger.

**SCBI vs SCBI with planning.** When preparing a curriculum, if the teacher does not plan ahead, they may base their teaching strategy for each round on Bayesian inference. This amounts to an initial teaching strategy of SCBI. However, if the same Bayesian teacher plans a curriculum which includes infinitely many data points, and uses the random variable $R(\theta) = \theta(h_{\text{target}})$ in their planning in order to effectively sample data points leading the learner to a stronger belief in the true hypothesis, the usual process of SCBI is recovered.

**Theorem 2** *Suppose that $\pi : \mathcal{D} \to \mathcal{H}$ is a teaching plan and $X$ is a reward/cost such that for every $d \in \mathcal{D}$, $\lim_{N \to \infty} \mathbb{E}_{\Psi_\pi^N \delta_{(d)}}[R]$ exists, and is independent of $d$. Then $\pi_R^\infty := \lim_{N \to \infty} \pi_R^N$ exists and is equal to the SCBI teaching strategy.*

**Proof:** Let $\theta \in \mathcal{P}(\mathcal{H})$, and $d \in \mathcal{D}$, and suppose that the teacher is teaching hypothesis $h \in \mathcal{H}$. To ease the readability, we will drop the subscript $R$ on the teaching plan. Then:

$$\lim_{N \to \infty} \pi^N(\theta)(d) = \lim_{N \to \infty} \frac{M_{(d,h)}^{\theta,\lambda} \mathbb{E}_{\Psi_\pi^N \delta_{(d)}}[R]}{\sum_{d'} M_{(d',h)}^{\theta,\lambda} \mathbb{E}_{\Psi_\pi^N \delta_{(d)}}[R]} = \frac{M_{(d,h)}^{\theta,\lambda}}{\sum_{d'} M_{(d',h)}^{\theta,\lambda}} = \frac{M_{(d,h)}^{\theta,\lambda}}{\theta(h)} \tag{10}$$

This is the formula used in the standard SCBI teaching strategy at each round of learning. $\square$

**Corollary 3** *With $\pi$ the standard SCBI teaching strategy and $R(\theta) = \theta(h)$, where $h$ is the target hypothesis, planning infinitely far ahead is identical to not planning ahead at all.*

**Proof:** By the proof of ((Wang et al., 2020a), Theorem 3.5), $\lim_{N \to \infty} \mathbb{E}_{\Psi_\pi^N \mu}[\theta(h)] = 1$, for any Borel measure $\mu \in \mathcal{P}(\mathcal{P}(\mathcal{H}))$. The claim then follows from Theorem 2 above. $\square$

The corollary above says that the limit of planning infinitely far ahead using SCBI is identical to SCBI with no planning ahead! Intuitively, this makes sense: in the infinite limit, the teacher is considering all possible infinite strings of data to pass to the learner; however, most of the strings will be indistinguishable as the learner approaches a particular belief on the true hypothesis, and so only the short-term behaviour of the strings is important. Furthermore, because the proof in Wang et al. (2020b) also shows that the convergence is monotone increasing; this implies that not planning ahead is the optimal teaching strategy when the reward at each round is equal to the probability that the learner would select the target hypothesis out of all hypotheses.

Figure 1 compares planning ahead up to four steps for two hypotheses and three data points, using $R(\theta) = \theta(h)$, with $h$ the hypothesis to be taught, assuming actions are selected probabilistically. The vertical axis is the learner's probability of choosing $h$ as the correct hypothesis if they were to randomly guess, while the horizontal axis represents the number of rounds of teaching/learning. The plot was created by randomly generating 1500 initial joint distributions, performing 50 rounds of teaching-learning with each 30 times, then averaging over the learner's performance.

An interesting feature of planning ahead with $R(\theta) = \theta(h_{\text{target}})$ is that if the teacher uses the deterministic procedure of argmax instead of sampling from the distribution to choose data to pass to the learner, the choice is the same for every number of steps planning ahead; i.e., in this case, local maximization is the same as global maximization (see Figure 1).

Other random variables are available to use in the expectation: for example, if there is one particular hypothesis which the teacher wishes to avoid, while biasing the learner toward the true hypothesis, the random variable could be $R(\theta) = 10 \cdot d_{L^1}(\theta, \delta_{h_{\text{bad}}}) - d_{L^1}(\theta, \delta_{h_{\text{true}}})$, where $d_{L^1}(\cdot, \cdot)$ is the $L^1$ distance between $\theta$ and $\delta_h$ represented as points of a simplex. In this case, there is non-monotonic behaviour, as the teacher overcompensates for trying to move the learner away from the 'bad' hypothesis, which subsequently may lead the learner closer to a neutral hypothesis than to the true hypothesis. See Figure (2), in particular the trajectories corresponding to planning ahead one step and two steps, where the probability that the learner selects the true hypothesis decreases.

For a comparison with a naive Bayesian learner, see Section 1 of the supplementary materials.

**Guaranteed alignment via monotonicity.** One of the key results of (Wang et al., 2020b) is the consistency of SCBI. Consistency here refers to the convergence in expectation of the learner's belief distribution to a Dirac distribution on the target hypothesis over the infinite limit of teaching rounds. In particular, Wang et al. (2020b) shows that if monotonicity holds, i.e. $\mathbb{E}_{\Psi_\pi \mu}[\theta(h)] - \mathbb{E}_\mu[\theta(h)] > 0$, where $h$ is the target hypothesis, then consistency and hence alignment follows. By writing out the equation above for monotonicity with $\pi$ replaced by $\pi_R^N$, for some choice of rewards/costs $R$, we obtain a condition for monotonicity and hence for alignment. Writing out the equation for $\mathbb{E}_{\Psi_{\pi_X} \mu}[\theta(h)]$ and $\mathbb{E}_\mu[\theta(h)]$, we may obtain an explicit condition for when $\mathbb{E}_{\Psi_{\pi_X} \mu}[\theta(h)] \geq \mathbb{E}_\mu[\theta(h)]$. Hence:

**Theorem 4** *Monotonicity holds if and only if for any $\mu \in \mathcal{P}(\mathcal{H})$:*

$$\int_\Delta \sum_d \pi_d(\theta) \cdot (T_d(\theta)(h) - \theta(h)) d\mu(\theta) > 0 \tag{11}$$

**Proof:** Throughout this proof, we rewrite $\pi(\theta)(d)$ as $\pi_d(\theta)$ for ease of readability. Expanding $\mathbb{E}_{\Psi_\pi \mu}[\theta(h)] - \mathbb{E}_\mu[\theta(h)]$ using the definition of $\Psi_\pi$ yields:

$$\int_\Delta \sum_d \pi_d \left( T_d^{-1}(\theta) \right) \theta(h) d(T_d^* \mu)(\theta) - \int_\Delta \theta(h) d\mu(\theta)$$

$$= \sum_d \int_{T_d^{-1}(\Delta)} \pi_d(\theta) T_d(\theta(h)) d\mu(\theta) - \int_\Delta \theta(h) d\mu(\theta)$$

$$= \int_\Delta \sum_d \pi_d(\theta) \cdot (T_d(\theta)(h) - \theta(h)) \, d\mu(\theta)$$

To get from the penultimate line to the final line, we use the fact that $T_d : \Delta \to \Delta$ is a homeomorphism of the simplex. $\square$

From this equation, we can see that if the $\delta_{M_{(d,-)}^{\theta,\lambda}}$ expectation of $R$ is overly negative for a hypothesis for which $T_d(\theta)(h) > \theta(h)$, while $T_{d'}(\theta)(h) < \theta(h)$ for other hypotheses, monotonicity will be broken. This implies that if the teacher places a heavy cost on a hypothesis lying close in belief-space to the target hypothesis, the curriculum of the teacher may over-emphasize moving away from the heavy cost hypothesis, at the expense of potentially converging to a more neutral hypothesis. Here two hypotheses $h_1$ and $h_2$ 'lying close in belief-space' means that with respect to the initial joint distribution $M$, the $L^1$ distance on the simplex $\mathcal{P}(\mathcal{D})$ between $M_{(-,h_1)}$ and $M_{(-,h_2)}$ is small. This implies that whatever data the teacher passes to the learner will affect both rows similarly.

## 4 Examples and simulations

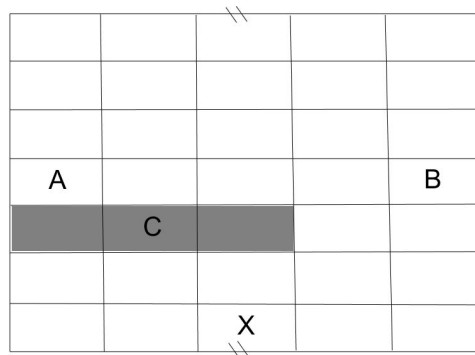 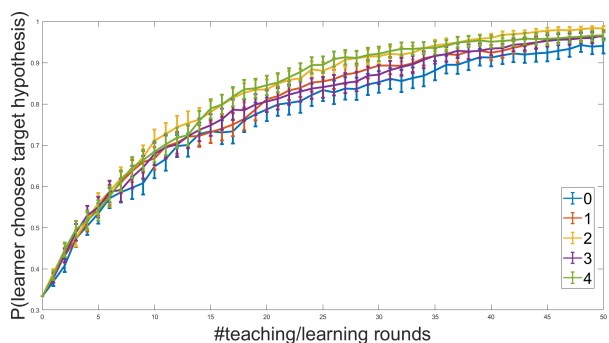

Figure 3: **(top)** GridWorld on a cylinder, with the entire top and bottom edges of the rectangle glued together. The grey region is hypothesis C, which is an undesirable hypothesis, and the target hypothesis is A. **(bottom)** Analysis of convergence to the target hypothesis, $A$, as a function of number of steps of planning ahead (0-4) for probabilistic action selection. Curricula yield marginal gains.

All simulations were run on a laptop computer with an AMD Ryzen 5 3550H with Radeon Vega Mobile Gfx 2.10 GHz processor or a macOS Mojave with 3.6 GHz Intel Core i7.

**GridWorld on a cylinder.** Suppose we have a grid on a cylinder, as in Figure 3 (left), where the top and the bottom edge have been glued together. In this problem, the teacher is attempting to direct an agent to the location $A$, while the location $C$, taking up three grid squares, represents a particularly undesirable location. In this case, the hypotheses correspond to the locations $A, B, C$, so $|\mathcal{H}| = 3$, and the data correspond to the four possible directions in the grid (up, down, left, right), so $|\mathcal{D}| = 4$. Starting from location $X$, the teachers who plan ahead out-perform the teacher who does not plan ahead on average, as shown in Figure 3 (right). When the teacher attempts to bias the agent away from $C$ and toward $A$ by using the random variable $R(\theta) = 10 \cdot d_{L^1}(\theta, \delta_C) - d_{L^1}(\theta, \delta_A)$ in planning, the agent is more likely to converge to the neutral

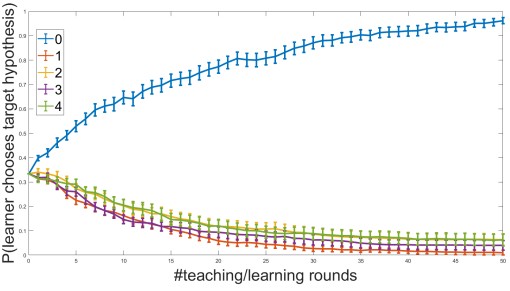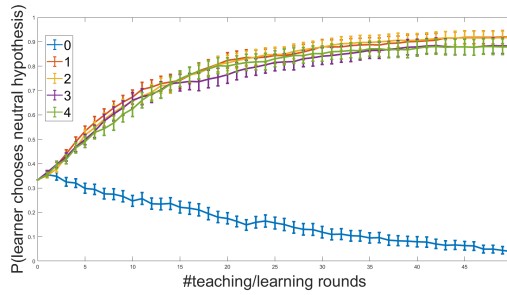

Figure 4: Analysis of convergence of the learner's beliefs for the random variable $R(\theta) = 10 \cdot d_{L^1}(\theta, \delta_C) - d_{L^1}(\theta, \delta_A)$ on the cylinder gridworld with a teacher that probabilistically selects data. **(left)** Probability that the learner guesses the correct location is $A$, the target hypothesis, versus # teaching/learning rounds. **(right)** Probability that the learner incorrectly guesses the location is $B$, the neutral hypothesis, versus # teaching/learning rounds. Avoiding the undesirable hypothesis, $C$, results in misalignment such that the teacher's examples lead the learner to the incorrect hypothesis $B$ when planning ahead.

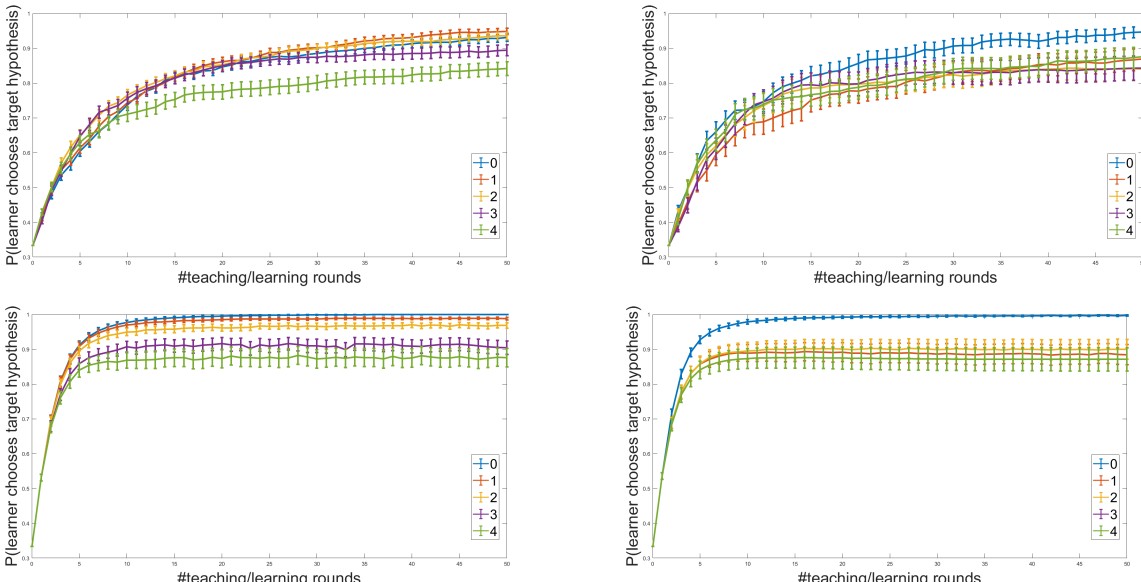

Figure 5: Analysis of the robustness of curriculum learning to errors in the teacher's estimation of the learner's prior, assuming reward $R(\theta) = \theta(h)$, with three hypotheses and four data points. Probability that the learner infers the correct hypothesis versus # teaching/learning rounds. **(upper left)** Average perturbation of size .001 on the teacher's estimation of the learner's prior, with the teacher sampling probabilistically. **(upper right)** Average perturbation of size .1 on the teacher's estimation of the learner's prior, with the teacher sampling probabilistically. **(lower left)** Average perturbation of size .001 on the teacher's estimation of the learner's prior, with the teacher using argmax to select data. **(lower right)** Average perturbation of size .1 on the teacher's estimation of the learner's prior with the teacher using argmax to select data. Longer curricula lead to greater misalignment due to propagation of error, and the misalignment is more severe with larger error.

hypothesis, as there is more incentive to move away from $C$ than to head toward $A$, so the agent gets closer to $B$. This is shown in Figures 4 (left) and (right).

**Robustness comparisons.** To compare robustness, we perturbed the teacher's estimation of the learner's prior at each round of teaching/learning. As shown in Figure 5, planning ahead is typically less robust than not. Each step of planning ahead relies upon estimating the learner's beliefs, which compounds the

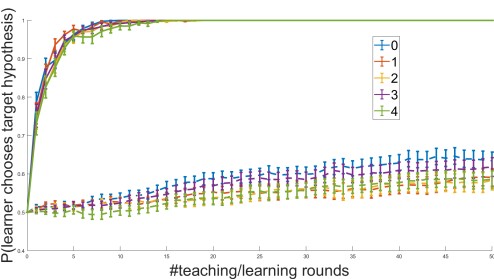

Figure 6: Convergence analysis when teacher accurately (solid lines) or inaccurately (dashed lines) estimates the learner's joint distribution when curriculum varies in the number steps planning ahead (0-4) with probabilistic data selection. Accurate estimates result in robust convergence across curricula. In contrast, inaccurate estimates result in severe alignment problems, which are worse for longer curricula.

overall error at each step of teaching/learning. In particular, Figure 5 shows that as the error increases, the convergence of the teachers who are planning ahead slows down, and that teachers who plan further ahead are less robust to perturbations in general.

As another test of robustness, we compare runs with two different initial joint distributions. This corresponds to the teacher and learner not having a shared knowledge of the relationships between data and hypotheses at the start. Another way to think of this is that one of the agents may have a mis-estimation of the relationship between data and hypotheses. Figure 6 shows that no tested strategies are robust with respect to an initial perturbation in joint distribution.

Section 2 of the supplementary materials contains simulations detailing convergence for various sizes of data/hypothesis spaces.

## 5   Related work

The mathematical theory of cooperative communication as optimal transport has been explored in (Wang et al., 2020b), (Yang et al., 2018), and (Wang et al., 2019). Sequential cooperation as utilized in this paper, including proofs of stability and rates of convergence, may be found in (Wang et al., 2020a).

Cooperative inference and reinforcement learning appear together in several contexts. For example, cooperative inverse reinforcement learning is centered around cooperative inference (Hadfield-Menell et al., 2016; Fisac et al., 2020), and cooperative inference and reinforcement learning often appear in the coordination of multiple agents, e.g. (Pesce and Montana, 2020). Milli and Dragan (2020) consider misalignment in terms of levels of theory of mind recursion. Our work differs in considering the problem of developing curricula to foster learning and offers novel theoretical and simulation-based insights into the possibility of misaligned curricula.

In curriculum learning (Elman, 1993), a learner is presented with carefully chosen examples or tasks, often increasing in complexity and carefully curated by the teacher. Such strategies are used in human cognition when teaching-learning complex tasks (Khan et al., 2011), and curriculum learning is a useful method in machine learning for gradually revealing complex concepts by slowly building up from simpler components (Bengio et al., 2009). Curriculum learning is used, for example, in neural networks in order to maximize learning efficiency (Graves et al., 2017), and multitask learning (Pentina et al., 2015). Curriculum learning has been included into the framework of a partially observable Markov decision process in Matiisen et al. (2020); Narvekar and Stone (2018). In Matiisen et al. (2020), the student is not directly observable by the teacher and the actions of the teacher correspond to teaching the learner on a certain task for a specific number of iterations. In Narvekar and Stone (2018), curriculum design is viewed as transfer learning, and the problem of learning a meta-policy is addressed. Our approach is distinctive in focusing on the alignment problem and offering strong mathematical proofs.

It is attractive to consider curricula that actively avoid common misconceptions. We are not aware of prior work in curriculum learning that does so, but it is common in reinforcement learning to consider worlds with negative reward, which is equivalent. In education, dealing with misconceptions most typically occurs through direct negation; for example, by including incorrect examples for students to correct (Heemsoth and Heinze, 2014) and through refutation texts (Tippett, 2010). Evidence suggests that such efforts are only effective when students know enough about the domain and are most effective at avoiding the misconception rather than inducing the correct belief. Thus, our findings regarding lack of monotonicity and misalignment are broadly consistent.

## 6 Conclusion

We investigate the possibility of alignment problems in curriculum learning by building recent theoretical advances in modeling cooperation. Recasting sequential cooperative Bayesian inference (SCBI) as a Markov decision process (MDP) enables the inclusion of curriculum learning to a teacher and learner cooperatively learning in multiple rounds of interactions. Through theoretical and simulation-based analysis, we show that curriculum planning introduces brittleness that leads to misalignment when the curriculum introduces additional costs, for example when avoiding a misconception, or when the teacher has imperfect knowledge of the learner. SCBI without curriculum planning offers competitive rates of convergence and desirable theoretical guarantees across our theoretical and simulation analyses. We also show that under simple assumptions on the reward/cost, e.g. taking the reward to be the probability that the learner will select the correct hypothesis, myopia appears to be optimal: planning ahead can actually decrease the rate of convergence, while converging to no planning as the number of steps ahead tends to infinity. Although this paper has primarily focused on the SCBI framework of cooperative communication, it is feasible that similar results exist for other MDP models of curriculum learning. Namely, naive choices of rewards/costs which may at face-value be reasonable, could result in misalignment or slower convergence, and are a potential pitfall for those working in reinforcement and curriculum learning. Future work should consider whether other curriculum learning approaches can offer similar theoretical and practical guarantees of alignment.

## Broader Impact

Our work is on alignment problems in curriculum learning. We have identified a new alignment problem which may be of longer term benefit if methods for addressing it are developed. There is no obvious reason why people may be put at disadvantage by this research, and it is unclear what the long term consequences of this work will be.

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
