# OpenReview forum: "The Alignment Problem in Curriculum Learning"
_TMLR — Withdrawn by Authors_

### Review · Reviewer_guFE · 2022-12-15

**Summary Of Contributions:**

This paper studies curriculum learning: in what order should data be presented to a learner to accelerate learning? The key idea is to view curriculum learning as a reinforcement learning (RL) problem. The paper studies this idea in a tabular setting, making certain assumptions on how the teacher and learner update their beliefs given the data seen so far. Under this model, the paper finds the doing multi-step reasoning/planning can be no better than acting myopically.

-----------------------------
**Update after reading the other reviews.**
I enjoyed reading the other reviews. I agree with Reviewer UFPm that these results seem to have interesting implications for the HRI community, while also agreeing that the lack of intuition (and notational complexity) will likely preclude these results from being recognized in that community. I agree with Reviewer 5t6n that some of the concerns raised in the previous reviewing round have not yet been addressed.

**Audience:**

Yes

**Broader Impact Concerns:**

This is a theoretical paper, without direct broader impact concerns.

**Claims And Evidence:**

Yes

**Requested Changes:**

The concern about motivation could be partially addressed by clarifying how the proposed analysis relates to prior frameworks for analyzing learning.

1.  How does this relate to online learning? There, it seems like the teacher is trying to _minimize_ learning; since existing online learning algorithms have good guarantees against this worst-case teacher, why do we need to design new learning algorithms that work with "easier" teachers?

2. Can we interpret the proposed method as "steering" the optimization path? What are settings where we care about the value of the intermediate points along that path, rather than the end point?

3. The RL objective in Eq. 8 seems intuitively similar to the statistical consistency and efficiency bounds used to analyze (say) MLE. Why is this RL objective preferrable over these prior notions of consistency?

One suggestion for improving the paper would be to have a running example (E.g., the learner is learning to do image classification). This would help clarify what the hypotheses are (e.g., sets of image classification models) and what the data are (e.g., pairs of images and labels).

Minor comments
* "one-off cooperation" -- In the abstract, I'd recommend providing a bit more explanation of what this means (e.g., add a sentence before this one to setup the problem). Giving some examples would help.
* "infinite length plans are equivalent to not planning ... planning ahead yields efficiency" -- At face value, this seems like a contradiction. I'd recommend addressing this (e.g., "While this may seem like a contradiction, ...")
* When introducing "data" in the notation section, it'd be useful to specify the "type" of the data. E.g., is this a set of (x, y) pairs?
* $\mathcal{D}$ isn't defined in the notation section.
* Background -- When reading this section, the question in my mind was why simple Bayesian filtering/updating wouldn't yield the desired results.
* Curriculum planning via MDPs -- When reading this section, I was wondering how this relates to prior work on learned optimizers?
* Eq. 9 -- There's a arg max over $d$ on both sides, but the expression doesn't seem to depend on $d$.
* Gridworld -- Again, it'd be good to clarify the "type" of the data in this example.

**Strengths And Weaknesses:**

Strengths
1. I think that the underlying problem is really interesting: when is long horizon planning/reasoning useful for learning tasks that otherwise would be challenging to learn?
2. It's great that this theoretical paper still contains some experiments to study the theoretical predictions.

Weaknesses:
1. I was a bit confused about the motivation. I'll try to explain the confusion. I tend to think about learning from a Bayesian perspective: as you observe data, you filter out models that are inconsistent with the data, leaving only those models that well-explain the data seen so far. Many ML models are exchangeable, in the sense that they assign equal likelihood to datasets regardless of the order in which that data are presented; e.g., a image classifier will assign the same log-likelihood to the ImageNet validation set regardless of whether the "dog" images are presented first or last. It seems like this paper is doing something different, assuming that the order in which data are presented affects the resulting models; I'm not sure this makes sense in most supervised and unsupervised learning settings. I think there might be a good argument for doing this for online, multi-task reinforcement learning: each round of learning would involve collecting data for some task, and that data might be useful for solving different tasks. However, this "RL to design a curriculum for RL" seems to be distinct from what this paper is discussing.
2. The introduction discusses claims that are not made in the later part of the paper. For example, the introduction starts by mentioning how "artificial systems may autonomously facilitate human goals including human learning," but the paper doesn't have any discussion of "human learning" (E.g., Do humans to SCBI?). Similarly, I'm not sure that the paper delivers on the promise to "bridge human and machine learning."
3. The notation is very complex. See, e.g., Eq. 5.

**Does the resubmission address the concerns raised in the previous submission?**
I don't think that the resubmission addressed the weaknesses raised in the prior review:
> 1. The paper doesn't include much discussion of whether the result generalizes to other formulations of curriculum learning
>
> 2. The notation is very hard to parse

---

### Review · Reviewer_5t6n · 2023-01-14

**Summary Of Contributions:**

The paper investigates the concept of Sequential Cooperative Bayesian Inference (SCBI), a student-teacher paradigm in which the teacher aims to convey a belief to a student by communication via data on which the student performs inference.

More precisely, the authors investigate whether SCBI, which has been previously modeled as a Markov Chain, benefits from a formulation as a Markov Decision Process in which an optimal teaching policy can be planned for a given reward objective.

Using the MDP formulation, the authors introduce a multi-step planning policy that uses the SCBI policy to plan multiple "teaching steps" ahead and investigate its behavior. They show that the multi-step planning policy converges to the SCBI policy in the limit of infinite planned teaching steps. Further, they investigate both empirically and theoretically how different reward functions can affect the consistency of learning. By doing this, they show that penalizing inference of undesired hypotheses can prevent the desired hypothesis from being learned.

**Audience:**

Yes

**Broader Impact Concerns:**

I do not have any concerns w.r.t. ethical implications of this paper.

**Claims And Evidence:**

No

**Requested Changes:**

The recommendations regarding the improvement of notation in my initial review stand basically unchanged. I see potential in this paper, but its impact is currently limited since it may be hard to digest by a wider audience.

With the inclusion of longer planning horizons and the visualization of the obtained rewards in Figure 2, the authors took a good step towards a more appropriate empirical verification of their theoretical insights. As outlined, I currently see evidence for issues in the computations/experiments and hence would like to see the authors carefully investigate those. This would either result in a clear description of why the observed artifacts are indeed no errors but in line with their theory and existing theory on RL, or an update of the experiments.

**Strengths And Weaknesses:**

In my [previous review](https://openreview.net/forum?id=tIdBsdyxG3&noteId=HCaPJDnJkW), my main concerns were focused on clarity of the paper due to sometimes unnecessarily complex notation, as well as the comparison of the approach to a properly computed optimal policy. I will go through the list of changes indicated by the authors and highlight how these changes affect my previous concerns:
* **Planning ahead as an approximation of the optimal policy has been emphasized:** I was not able to find an additional emphasis on an optimal policy in the paper. I contrasted the new submission to the latest revision of the rejected paper. Could the authors please clarify where to find this additional emphasis?
* **On page 2, in the 'Notation' section, 'row' and 'column' have been interchanged** + **In Equation 3, an intermediate equation with the same meaning but more confusing notation was removed:** While any fixes of typos and improvements in notation are welcomed, these two changes are very minor and do not address the (in my opinion) unnecessarily complicated notation for e.g. the MDP transition dynamics (see my initial review for a more detailed discussion).
* **Figure 2 has been updated to include the maximum number of planning ahead steps (seven steps ahead):** The change in Figure 2 is highly welcomed. However, the additional visualization of the achieved reward raises multiple questions on my side (ordered from most important to least):
    * The rewards of the "planning-ahead" policies raise doubts on the correctness of the results. For any MDP, an optimal policy with a planning horizon of N time-steps should achieve the highest possible cumulative reward over these N time-steps. However, the "no-planning" policy achieves a significantly higher reward for N=1-7. This sub-optimality of the planned policy indicates that there is either a mismatch between the MDP used for planning and for evaluation, or that the optimal policy is not properly computed.
    * The maximum immediate reward is 10 but the maximum values visualized in Figure 2 take on values of 20. I hence assumed that the Figure shows the cumulative reward subject to discounting instead of the immediate reward. However, this assumption is again contradicted by the reward curves starting at a value of 12, which is higher than the maximum possible immediate reward of 10. Can the authors clarify what rewards are shown in Figure 2?
    * The reward definition in the caption of Figure 2 and on page 7 seems to contain a typo. Shouldn't it be $R(\theta) = 10 d_{L_1}(\theta, \delta_{h_{\text{true}}}) - d_{L_1}(\theta, \delta_{h_{\text{bad}}})$? Currently, the reward as defined in the paper increases, if the teacher conveys the bad hypothesis.
* **In Figure 3, the dimensions of the subfigures have been adjusted in order to make them more readable:** Again, this change is welcomed but arguably minor. It does not address the main concerns without using the additionally generated free space for additional clarifications.

---

### Review · Reviewer_UFPm · 2023-01-17

**Summary Of Contributions:**

This paper examines the problem of sequential cooperative inference, wherein a teacher presents data to a learner in order to teach the learner a hypothesis, and re-casts the problem in the context of RL where the aim is to get the learner to identify a goal state. The paper presents theoretical results regarding the conditions for alignment (ie., where the learner successfully recovers the hypothesis) and examines the effect of planning (or not) by the teacher. The paper demonstrates that not planning at all is equivalent to planning infinitely far in the future, and that when using short-term planning, certain reward functions can actually result in misalignment between the teacher and learner.

**Audience:**

Yes

**Broader Impact Concerns:**

No concerns.

**Claims And Evidence:**

Yes

**Requested Changes:**

1. My biggest question/concern is around the intuition for some of the results. Despite reading through the methods several times, I'm still unclear on what the explanation is for why planning doesn't help, and in some cases, hurts. Some thoughts:
  - I guess the result that planning infinitely long is equivalent to not planning at all, can be explained by saying that the SBCI teaching strategy with $R(\theta) = \theta(h)$ is already the optimal policy for the teacher, so planning cannot improve on it. It would be nice if you could say this explicitly. I think this is implied, by presenting first the optimal policy as a function of $N$, and then showing that as $N\rightarrow \infty$ you recover the SBCI teaching strategy. But there are two things missing: first, explicitly saying "and therefore the SBCI teaching strategy *is* the optimal policy". Second, it would be helpful to also show that planning with $N=0$ is equivalent to SBCI as well.
  - I still don't fully understand the reason why planning would hurt for short horizons. I guess the point is that this is only true when you have a reward function that penalizes certain belief states, and by planning the teacher believes that the learner will pass through/near these belief states, and so decides to give it different data, which ends up steering it too far away from the correct hypothesis. However, when planning for an infinite horizon, the teacher would realize that the learner ends up in the wrong place and so avoids this strategy. Is that intuition correct? If so, it would be helpful to add this intuition to the paper. If not, could you please clarify (and also add the explanation to the paper).
2. My next concern is about some of the assumptions: that the teacher can perfectly estimate the learner's belief state, and that the learner is an optimal Bayesian learner. Both of these seem quite strong and that they would not generally hold in practice. Moreover, the results in Figure 6 suggest that violating the first assumption is a big problem (though it's unclear to me how big of a perturbation there was in generating this Figure). It would be helpful to provide some discussion of this, and even better if you could justify why these are reasonable assumptions (e.g. are there some real-world settings for which they would hold?).
3. Finally, I am not an expert in ML/probability theory, and I found the math to be quite dense and difficult to parse. This made it difficult for me to evaluate the correctness of the math. In several cases the paper seems to rely on mathematical concepts that I don't think are common knowledge for all ML researchers, such as concepts from measure theory. This is a shame, because I believe it limits the accessibility of the paper to researchers from areas such as HRI/HCI where it is very relevant but where researchers might not be familiar with the terminology. It would be helpful to provide more accessible explanations and perhaps even leave some of the nitty-gritty details to the supplementary material.

Other questions/concerns:
- The abstract says "we prove that infinite length plans are equivalent to not planning under certain assumptions on the method of planning", but as far as I could tell there was not any discussion of the planning method in the main text. Indeed, the fact that there was no discussion of the method of planning feels like a shortcoming. I assume you are using something like a breadth-first search in the experiments but it would be good to state this explicitly.
- The abstract says that "planning ahead yields efficiency at the cost of effectiveness". What does efficiency mean in this context? There is no discussion of efficiency in the main text. Do you mean data efficiency, wall clock time, etc.? But for any of these choices, it seems like your results do not suggest planning is more efficient? So I am confused by this statement.
- I am not sure what exactly Figure 2 (lower) corresponds to, is it the reward for selecting actions deterministically or probabilistically? The caption could be clearer in this regard.
- Please include details of the experiment shown in Figure 6. What exactly does the "inaccurate" estimate refer to? How is the joint distribution perturbed, exactly?

Smaller issues:
- The font for the tick labels in the figures should be larger.
- argmax is rendered incorrectly in Equation 9
- In Theorem 2, X does not seemed to be used. Should it be R?

**Strengths And Weaknesses:**

Pros:
- studies an important problem in multi-agent RL and human-agent interaction
- clear theoretical results plus simulation results

Cons:
- lack of intuition provided for some of the results
- the results are dependent on some strong assumptions, such as having perfect knowledge of the learner and that the learner is optimally Baysian
- mathematical notation is difficult to parse
- some details missing

---

### Review · Reviewer_Yzyk · 2023-01-23

**Summary Of Contributions:**

This paper investigates an extended version of the problem setting of Sequential Bayesian Cooperative Inference (SBCI), in which a teacher and a student model each other's beliefs based on shared knowledge of a joint distribution over a finite number of hypotheses and data points. Specifically, the authors extend this setting to include (1) the notion of a reward/cost function (of the hypothesis distribution of the student at each time), and (2) the ability to plan ahead for some (potentially infinite) number of steps. The authors prove theoretical properties of this setting, notably a monotonicity condition that guarantees the student converging to the intended (or "aligned") hypothesis, and (2) showing that when the reward at each time is equal to the probability of the student choosing the correct hypothesis, planning with an infinite time horizon is equivalent to no planning. The authors also present simulations supporting their theoretical conclusions.

**Audience:**

Yes

**Broader Impact Concerns:**

No concerns.

**Claims And Evidence:**

No

**Requested Changes:**

- Given the large gap between the finite SCBI setting studied and the motivated general setting of a teacher agent faciliating the learning of a student agent, I think the authors should change the title of the paper to be less general and more specific to the setting of study, e.g. “The Alignment Problem in Cooperative Inference.”
- The discussion of prior work on curriculum learning feels a bit one-dimensional. It only mentions a few, now slightly outdated results, without mentioning more recent works around both minimax and minimax-regret curricula that are theoretically grounded in decision theory (see list of suggested references). This makes the comment “Theoretical results are limited” in the introduction inaccurate:

    *Example of minimax curricula for robust RL*************************************************************:************************************************************

    - Pinto, Lerrel, et al. "Robust adversarial reinforcement learning." *International Conference on Machine Learning*. PMLR, 2017.

    *Examples of minimax-regret curricula for zero-shot transfer in RL*****************:****************

    - Dennis, Jaques, et al. "Emergent complexity and zero-shot transfer via unsupervised environment design." *Advances in neural information processing systems*
     33 (2020): 13049-13061.
    - Jiang, Grefenstette & Rocktäschel. "Prioritized level replay." *International Conference on Machine Learning*. PMLR, 2021.
    - Jiang, Dennis et al. "Replay-guided adversarial environment design." *Advances in Neural Information Processing Systems*
     34 (2021): 1884-1897.
    - Parker-Holder, Jack, et al. "Evolving Curricula with Regret-Based Environment Design." PMLR 2022.
    - Lanier et al. "Feasible adversarial robust reinforcement learning for underspecified environments." *arXiv preprint arXiv:2207.09597*
     (2022).
- Further, the introduction mentions that prior works “have not systematically considered the possibility of alignment problems or their solutions.” This is also factually inaccurate. Two works that consider this in both the deep RL setting and the supervised learning setting are referenced below:
    - Jiang, et al. "Grounding Aleatoric Uncertainty for Unsupervised Environment Design." NeurIPS 2022.
    - Kirsch, Rainforth & Gal. "Test Distribution-Aware Active Learning: A Principled Approach Against Distribution Shift and Outliers." *arXiv preprint arXiv:2106.11719*
     (2021).
- Moreover, curricula are used for many different purposes (e.g. to improve transfer to a known target task as in the work of Narvekar et al; to improve sample efficiency as in Matiisen et al; to improve adversarial robustness as in Pinto et al; and to improve zero-shot transfer to unknown test-time tasks as in Dennis et al). Thus, the authors should avoid referencing prior work in curriculum learning as a monolithic body of work.
- At a high level, these requests for changes stems from the paper overclaiming the relevance and contributions of the study presented. The contributions of the paper would be better conveyed if the authors contextualized their specific setting more precisely with respect to prior works and grounded their discussion in the context of their specific problem setting.

**Strengths And Weaknesses:**

### Strengths

- This paper looks at an important problem in systems in which agents facilitate the learning of other agents, e.g. curriculum learning.
- The paper provides theoretically-grounded conclusions about a specific setting of cooperative inference.
- The paper provides experimental results that largely support the theoretical conclusions.

### Weaknesses

- The largest weakness of this work is the disconnect between the problem setting studied and the broader problem setting initially motivated in the introduction, which is the setting of general teacher-student curricula, e.g. in supervised and reinforcement learning. The problem setting studied is limited to small, finite tabular SCBI problems. The extension of the theoretical results and algorithms studied in this setting to deep supervised or reinforcement learning (RL) settings, as mentioned in the motivating passages, remains unclear. Such an extension seems nontrivial as in the more general supervised and reinforcement learning settings (over the kinds of POMDPs studied in the deep RL literature), there is not a finite hypothesis space, and generally neither the teacher nor student have access to the joint distribution of hypotheses and data. **It is thus unclear whether any of the theoretical insights regarding the extended SCBI setting studied in this work carries over to the motivated problem setting. Therefore, I believe the current framing of the paper to overclaim the significance of its contributions.**
- The implications of the experimental results are not clearly communicated in the context of the theoretical contributions. In parts, the results also seem somewhat inconsistent, and this inconsistency is not fully explained: For example, in Figure 1, the caption states that “On average…longer curricula yield a marginal difference,” but the corresponding plot shows that the probability of the student choosing the target hypothesis seems to decrease with a longer planning horizon. This marginal difference is first explained as planning not making a large difference when the reward at each time is equal to the probability of the learner selecting the target hypothesis. However, later in Section 4’s GridWorld results (Figure 3), the performance (in terms of probability of learner choosing target hypothesis) seems to consistently increase with the planning horizon.
- Throughout the paper, the theory can be better connected to easier-to-grok intuitive explanations. This would be particularly helpful for some of the concepts first defined in the Notation section, e.g. Sinkhorn normalization, as well as optimal transport. Intuitive grounding of these concepts and how it relates to the SCBI setting studied would benefit the work, as these concepts seem to straddle separate and somewhat disjoint communities of research interests (which on the flipside, is an aspect of the work that makes it interesting).
- The “shared joint distribution” aspect of the SCBI setting studied should be emphasized and clearly stated upfront.

---

### Author Response · Authors · 2023-02-10
**Thanking the Reviewers**

We wish to thank the reviewers and action editor for their time and for their useful comments. Unfortunately we feel unable to adequately address the concerns of the reviewers before their decision, due to the tight turn-around time. As such, we will withdraw our submission. Thank you again for your time and for your reviews.

---

### Note · Authors · 2023-02-10

**Comment:**

We wish to thank the reviewers and action editor for their time and for their useful comments. Unfortunately we feel unable to adequately address the concerns of the reviewers before their decision, due to the tight turn-around time. As such, we will withdraw our submission. Thank you again for your time and for your reviews.

**Withdrawal Confirmation:**

I have read and agree with the venue's withdrawal policy on behalf of myself and my co-authors.